# Intercropping Systems to Modify Bioactive Compounds and Nutrient Profiles in Plants: Do We Have Enough Information to Take This as a Strategy to Improve Food Quality? A Review

**DOI:** 10.3390/plants13020194

**Published:** 2024-01-11

**Authors:** Ana Patricia Arenas-Salazar, Mark Schoor, Benito Parra-Pacheco, Juan Fernando García-Trejo, Irineo Torres-Pacheco, Ana Angélica Feregrino-Pérez

**Affiliations:** Faculty of Engineering, Autonomous University of Querétaro, Cerro de las Campanas s/n, Querétaro 76010, Mexico; arenas.uaq@gmail.com (A.P.A.-S.); schoor.uaq@gmail.com (M.S.); benito.parra@uaq.mx (B.P.-P.); fernando.garcia@uaq.mx (J.F.G.-T.); irineo.torres@uaq.mx (I.T.-P.)

**Keywords:** food production, sustainable agriculture, human health, nutrients, bioactive compounds

## Abstract

Various environmental, food security and population health problems have been correlated with the use of intensive agriculture production systems around the world. This type of system leads to the loss of biodiversity and natural habitats, high usage rates of agrochemicals and natural resources, and affects soil composition, human health, and nutritional plant quality in rural areas. Agroecological intercropping systems that respect agrobiodiversity, on the other hand, can significantly benefit ecosystems, human health, and food security by modifying the nutritional profile and content of some health-promoting bioactive compounds in the species cultivated in this system. However, research on intercropping strategies focuses more on the benefits they can offer to ecosystems, and less on plant nutrient composition, and the existing information is scattered. The topic merits further study, given the critical impact that it could have on human nutrition. The aim of this review is therefore to collect viable details on the status of research into the profile of nutrients and bioactive compounds in intercropping systems in different regions of the world with unique mixed crops using plant species, along with the criteria for combining them, as well as the nutrients and bioactive compounds analyzed, to exemplify the possible contributions of intercropping systems to food availability and quality.

## 1. Introduction

Global population growth and lifestyle changes have increased worldwide food demand [1,2]. Intensive food production systems like monoculture have been implemented to meet this demand. However, intensive agriculture has hurt the environment, causing a loss of biodiversity and natural habitats and making excessive use of agrochemicals and fertilizers, which leads to the loss of fertile soils by damaging the soil microbiota (plant-soil interaction) [3,4], while the contamination of aquifers affects the availability of nutrients required by crops [5,6,7]. Furthermore, intensive agriculture systems have yet to prove themselves capable of eradicating hunger in developing countries; even worse, this group of faulty agricultural practices results in low nutritional quality in the products intended for human and animal consumption, which could be related to health issues [8]. Thus, various food production strategies must be tested to guarantee a more adequate food supply. It has been stipulated that balancing biodiversity conservation and food security is the key to global sustainable development [3,9]. In this context, sustainable agriculture integrates multiple areas: it addresses environmental, market, policy, research and innovation concerns, along with several societal benefits [10], like improving the status of human health [11] by improving food quality [12]. In principle, the food produced under sustainable agriculture systems would be more accessible, affordable, safe, and equitable, which also meets the requirements of food security [13,14].

Intercropping systems are traditional farming practices in which two or more crops are planted together, through seed mixing or through various spatial arrangements, on the same land at the same time [15]. They may provide a viable form of sustainable food production thanks to their diversification [16]. What these systems purport to do is to achieve an interaction that benefits the different species involved and provides more regulation services to ecosystems [17,18]. Intercropping systems could have a positive impact on the environment and society by preventing soil erosion and improving soil fertility through enriched soil microbiota. Moreover, they can increase the biodiversity and conservation of natural habitats due to the different families of plants grown within the same area, which provide a natural habitat to many species, and naturally regulating pests, diseases, and weeds could considerably reduce the use of fertilizers and agrochemicals [18].

Furthermore, farmers may also benefit from socioeconomic gains and greater food security, since multi-cropping can improve yield [19], which would increase the availability of food or allow them to sell more of their produce [20,21,22]. All these characteristics have been widely studied over the years; nevertheless, few studies have analyzed the changes in the quantity and quality of nutrients and phytochemicals that benefit human health due to the interaction of the species involved in intercropping. One example is the amount of soluble and insoluble fiber, increased phenols and flavonoids, amino acids, and other phytochemicals found in fruits and/or flowers [9,23,24]. The study of these possible changes is relevant for food and nutritional security since, for developing countries, intercropping systems could supply a large portion of families’ nutritional needs [14].

To our knowledge, no study has yet compiled the research carried out on intercropping to identify changes in the quantity and quality of bioactive compounds beneficial to human health or macronutrients. Therefore, the objective of this review is to collect viable information about the current status of research in various regions of the world where intercropping systems are in use for nutritional improvement purposes, the species that have been used in these crops, and the criteria for combining these species, and the nutrients that were analyzed.

## 2. Food Security and Intercropping Systems

In recent years, there has been greater awareness of the challenges and actions needed to eliminate hunger and malnutrition worldwide, including food system management. In this sense, agriculture plays a vital role in most developing countries. However, due to population growth, industrial development, and political factors, current food systems have proven insufficient to feed this growing population [2]. Despite the widespread use of intensive food production systems, the goal of eradicating world hunger has not been achieved [25]. Indeed, around the world, approximately 2.37 million people experienced food insecurity at a moderate or severe level in 2020 [26].

### 2.1. Food Security and the Importance of Sustainable Production

Food security is a term that was first introduced in 1970, and it has been successfully redefined to adapt to the needs of the current global population [27]. All existing reports generally agree that food security is defined as the provision of a sustained food supply [28]. The World Food Summit added that food security focuses on four main dimensions: availability, accessibility, utilization, and stability. These four dimensions do not necessarily coincide. For this reason, food security cannot be adequately measured with a single indicator. A multidimensional analysis is needed to assess and compare various food security indicators at the regional and national levels [29].

A widely accepted definition of food security, supplied by the Food and Agriculture Organization (FAO), starting in the 1990s, was “a situation that exists when all people, at all times, have physical, social and economic access to sufficient, safe and nutritious food that meets their dietary needs and food preferences for an active and healthy life” [30]. This was the first time that emphasis was placed on the importance of nutrient quality contained in foods for human consumption, considered crucial to the dimension of utilization [27]. However, it is also known that the nutritional quality of foods produced in intensive systems has decreased, causing a deficiency in the micronutrients available to the population [31]. Evidently, the “utilization” dimension is not fulfilled by current food production systems.

Food security also considers how food production systems perform in terms of optimized cultivation intensity. However, growing environmental damage has changed how this indicator is measured in evaluating food security. More recently, the food security framework has incorporated two more dimensions: agency and sustainability [27]. Agency involves compliance with policies related to food security, especially when establishing political frameworks and institutions to defend the rights of the most vulnerable groups. Sustainability requires that food systems are appropriately managed to contribute to the long-term and present-day regeneration of natural, social, and economic systems, so that food sufficiency can be ensured for current and future generations [32].

Sustainability is a core component of the UN Sustainable Development Goals (SDGs). In 2021, it was a central objective of the United Nations Summit, where Food Systems issues were discussed. At this meeting, it was emphasized that healthier, more sustainable, and equitable food production systems are required worldwide. The search for new and better food production systems has become a global priority.

One of the systems proposed in recent years is intercropping, a traditional farming system [33] that is considered to be a diversified and sustainable agricultural technique that optimizes cropping intensity.

Many studies show that intercropping can offer different ecosystem services [34] because inputs and natural resources are better used to supply nutrients that protect plants against pathogens, pests, and weeds. It can also improve soil fertility, conserve biodiversity and natural habitats, and provide higher yields and more balanced production per unit area due to crop diversification [20,35]. Further research is necessary, but it has been suggested that the types of plants included in an intercropping system could increase or decrease the chemical composition of some bioactive compounds and nutrients. The health benefit is related to the possible nutraceutical effect that plant-based foods could obtain during their development in the crop [23,36,37]. This may provide the consumer with a greater quantity and quality of nutrients, thus helping to prevent illness or improve health [38], which would be especially beneficial to communities where food security and health status are currently compromised.

Therefore, implementing sustainable agroecological systems like intercropping could have a positive environmental impact and produce healthier food, which coincides with food security policies and the SDGs.

### 2.2. Food Quality and Human Nutrition Related to Food Production Systems

One of humanity’s perennial challenges is developing food production systems that can ensure food security.

Health status has been found to relate closely to diet and, therefore, to people’s nutrition. At present, nutrition is affected by many factors, including changes in people’s lifestyles, and everything else that comprises food security [11]. This is why, especially in developing countries, various health problems have emerged in connection with malnutrition [39,40]. These problems are attributed to the tendency to consume high quantities of fats, sugars, and processed foods with a high caloric density and low dietary fiber and water, since these tend to be foods that are economically accessible to the population. At the same time, the consumption of fruits, vegetables, and complex carbohydrates from essential food production in these developing countries is decreasing at an alarming rate [41,42]. This food transition also entails an epidemiological transition, with the rising incidence of different diseases related to these habits: cardiovascular diseases, various types of cancer, hypertension, type 2 diabetes, polycystic ovary syndrome (PCOS), stroke, and many others associated with overweight and obesity [43,44,45,46]. In general, the population has been told that the sufficient and varied consumption of unprocessed foods and lifestyle changes can support the prevention of these diseases. For small farmers in developing countries, this variety of food could be obtained from subsistence agriculture, so their food and nutritional security depends on this. Nevertheless, the need to produce more food grows each year and has been the motive for the deployment of intensive monoculture systems around the world. At present, there are growing concerns about not only the quantity of food these systems produce, but also the nutritional quality of that food [31]. The indiscriminate use of monoculture has caused a significant deficit of micronutrients in the edible parts of food crops, which means that the population cannot be adequately nourished even by consuming unprocessed foods, nor can they avoid chronic degenerative diseases. This may ultimately result in a phenomenon known as hidden hunger [31], defined as a dietary deficit in the intake of vitamins and minerals, such that the food consumed is inadequate for optimal human health. In recent years, there has been a troubling increase in the prevalence of disease and illness in lower-income countries. The World Health Organization (WHO) estimates that, in 2023 [47], more than a quarter of the global population will suffer from one or more micronutrient deficiencies. The most common deficiencies of micronutrients registered are vitamin A, iron (Fe), zinc (Zn), and iodine (I) [48]. Micronutrient deficiencies could be a risk factor for many diseases, because without them the organism becomes less resistant to infections that cause severe illnesses, including anemia, mental retardation, blindness, and spinal and brain birth defects [49].

Another critical aspect of food security is guaranteeing the safety of the ingested produce and the entire process of obtaining that food. The first studies on the interaction between agrochemicals and human health are beginning to emerge in different countries, particularly the effects of the intensive application of pesticides [50]. Previously, the damage that exposure to these chemicals could cause to human health was unknown; however, new scientific evidence has alerted the population about the indiscriminate use of these substances. The symptomatology of acute poisoning due to excessive use of phytosanitary products such as fungicides and bactericides, herbicides, and insecticides, among others, may be well known, but the subclinical consequences related to prolonged exposure to these agrochemicals have been little studied. Few studies relate these chemicals to cognitive impairment, reproductive disorders, cancer, diabetes, neurobehavioral and neurodevelopmental disorders, congenital malformations, and cardiovascular, respiratory, and neurodegenerative diseases, such as Parkinson’s and Alzheimer’s [51,52,53]. In addition, the use of agrochemicals is poorly regulated and controlled in some countries, which implies a greater risk for the population [50].

These health problems could be solved by restoring the diversity of agricultural ecosystems, managing crops effectively, and limiting deleterious environmental effects. If sustainable production systems are correctly implemented, the produce obtained will be of greater nutritional quality.

## 3. Modifying the Nutritional Profile in Intercropping Systems

Backyard production is a system for managing plants in traditional land use systems in areas close to homes in developing countries [54]. Promoting the creation of such spaces allows families to cultivate multiple species to cover their basic needs [55], providing most of their necessary daily nutrients. In situations of food scarcity, it is a critical practice [56,57]. Plant seeds and other edible plant parts are food sources that are rich in essential nutrients: lipids, peptides/proteins, amino acids, starch, dietary fiber, vitamins, and minerals [58]. Likewise, some bioactive compounds are derived from the plant’s seeds, fruits, roots, and leaves; these are phytochemicals like phenolic compounds (tocopherols, flavonoids, and phenolic acids), nitrogen compounds (alkaloids, chlorophyll derivatives, amino acids, and amines), carotenoids, or ascorbic acid, quinones, terpenoids and saponins [59,60].

In recent years, consumer preferences have shifted toward sustainable food production, similar to agricultural techniques in family gardens [61]. Interest in food quality, functional foods, and eating seasonally, locally, and organically has been growing [12,62,63]. Accordingly, the search for crops with high nutritional content has been a challenge worldwide [64]. The “sustainable diet” is a term that was established with sustainable food production in mind; this concept includes all dimensions of people’s health and well-being because it has a low environmental impact and is accessible, affordable, safe, and equitable, meaning it also conforms to requirements of food security [13]. Moreover, sustainable diets can preserve traditional regional cuisine as a part of the intangible heritage of societies and communities, and can play a key role in regional and local economies [65].

Mixed cropping, as a form of sustainable agriculture, can help low-income households to afford a more diverse diet, improving their daily intake of essential foods. Combinations of cereals, legumes/seeds, and oilseeds in intercropped systems can provide a large part of families’ caloric intake [14]; therefore, these systems can play a vital role in alleviating hunger, especially if they are implemented in a manner that augments the content of nutrients and bioactive phytonutrients [64]. Nevertheless, few studies have been conducted on the introduction of intercropping systems with an eye to improving the nutritional quality of the cultivated species.

The existing studies on the nutritional quality and quantity of bioactive compounds and macronutrients that can be modified using intercropped systems find that a number of countries have implemented these systems according to their dietary needs and species of interest. It is interesting to analyze the methodologies used in these studies and the type of nutrient that was modified. Food quality is known to depend on genetic factors, environmental conditions, growing location, and agronomic practices. Table 1 combines the most relevant characteristics of various investigations where an intercropped system was implemented to modify the nutritional profile of one or more involved species to benefit human nutrition.

In this context, there are studies that emphasize the modification of bioactive compounds, while others focus on the change in the quantities of other nutrients, and others search both profiles.

### 3.1. Intercropping Systems with Cereals and Legumes

In general, the intercropping of cereals and legumes is a global practice. It has been widely used to increase crop yield due to the biological nitrogen (N) fixation. In recent years, various investigations have discovered that this system may induce an increase in crude protein in one or both species. This improvement is related to N and phosphorus (P) transfer from the legume to cereal during their co-growing period in these intercropping systems [66,67,68,69].

**Table 1 plants-13-00194-t001:** Research where an intercropping system was introduced to modify both bioactive compounds and macronutrient content (or one of them) of one or more of the cultivated species, for the benefit of human nutrition.

Species Involved	Methodology	Bioactive Compounds	Macronutrients	Country or Climatic Zone	Author/Year
Maize and peanut Maize and soybean	Six treatments with two intercropping systems, maize–pea and maize–soybean, with and without the application of fertilizer and their respective monocultures; each treatment was replicated three times. Plot area: 33 m^2^ (6 m × 5.5 m); the field experiment had a total of 18 plots.	Maize intercropping (peanut and soybean) increased the lysine content of maize grains when no fertilizer was applied. When fertilizer was applied in both intercropping systems, the content of lysine increased.	Maize intercropping (peanut and soybean) significantly increased the protein and oil content of maize grains when no fertilizer was applied. When fertilizer was applied in both intercropping systems, the content of starch increased.	China	[70]
Barley and alfalfa	Intercropping pot experiment with AMF and PGPR. Three inoculation treatments (for both monocropped and intercropped plants) and the control were used: (1) AMF-inoculated plants; (2) PGPR-inoculated plants; (3) AMF + PGPR co-inoculated plants	Intercropping and co-inoculation of AMF + PGRPR increased the total phenolic 132%, and flavonoid 343% content of barley grains.	Intercropping and co-inoculation of AMF + PGRPR increased protein in 99%.	Marrakesh, Morocco	[71]
Wheat and faba bean	Intercropped wheat and faba bean with (N) fertilization: N0, no N fertilizer applied to both wheat and faba bean. N1, 90 and 45 kg N ha^−1^ applied to wheat and faba bean. N2, 180 and 90 kg N ha^−1^ applied to wheat and faba bean. N3, 270 and 135 kg N ha^−1^ applied to wheat and faba bean. Control group: Wheat and faba bean monoculture.	--------------------	Wheat grain protein content increased by 9% with N3 level, NEAAs content was 31% higher under the N1 level and, grain EAAs was increased by 39% at the N1 level relative to monoculture wheat.	China	[69]
Spring wheat and different legumes	Comparison of two systems—mixture and row-by-row cropping—in 3 different locations.	-------------------------	The row-by-row cropping system resulted in higher crude protein content (14.02%) than the mixture (13.79%). Zvhad (Zv) had the highest crude protein content (15.14%).	Czech Republic; Prague (PR), Uhříněves (UH) and Zvíkov (Zv).	[72]
Wheat and clover	2 types of trials: “Broadcast” with three treatments: 1. Unfertilized system, where wheat was sown in paired rows (330 seeds m^2^, 21%) and clover was broadcast sown (1250 seeds m^2^, 79%) (Pcwbc); 2. Unfertilized wheat as a sole crop, sown in paired rows (330 seeds m^2^) (Ctrlpr); 3. Wheat as a sole crop, sown in single rows (440 seeds m^2^), and fertilized with organic poultry manure (Ctrl). “Row” trial with three treatments: 1. Unfertilized system, where both wheat (330 seeds m^2^, 21%) and clover (1250 seeds m^2^, 79%) were sown in paired rows (Pcw); 2. The Ctrlpr (330 seeds m^2^), as in the “Broadcast” trial. the Ctrl (440 seeds m^2^) treatments, as in the “broadcast” trial.	--------------------------	Wheat grain protein content was 16% and 24% higher in Pcw and Pcwbc, respectively, than in Ctrlpr, and 15% and 28% compared to Ctrl.	Region around Pisa, Italy (sites: Valtriano and Santa Luce	[73]
Milpa (colored corn, climbing bean, and squash, tobacco) with potato, 3 classes of peppers—namely poblano, jalapeno and bell—, beetroot, carrot and kale.	All vegetables were first grown in greenhouse, except potato tubers, which were directly planted in the garden plots. Forty-five-day-old seedlings were transplanted at Probstfield Organic Community Garden. No chemical fertilizers were used for this study.	Kale had the highest total soluble phenolic (TSP) content with 1.02 mg/g FW. It also had the highest phenolic acid content, detecting dihydroxybenzoic acid, ferulic acid, and cinnamic acid. Two of the three classes of peppers had higher concentrations of phenolic acids: jalapeno (gallic acid and p-coumaric acid) and poblano (benzoic acid, dihydroxybenzoic acid, and catechin).	------------------------	Northern plains USA	[74]
Fenugreek seeds with buckwheat	Two-year experiment with four treatments: Sole fenugreek (control) with three intercropping ratios with buckwheat; F:B = 2:1, 1:1, and 1:2 each with three types of fertilizer (chemical fertilizer, integrated fertilizer, and broiler litter. Researchers studied trigonelline content, antioxidant activity measured with DPPH and FRAP, total phenolic and flavonoid content, and specific flavonoid contents of fenugreek seeds.	Results in intercropped fenugreek seeds: -Antioxidant activity: Higher DPPH levels, on average, by 12.3% (2014) and 12.5% (2015) compared to Sole F, thus increased antioxidant activity. The highest antioxidant activity was measured in the F:B = 2:1 plots with 4.25 (2014) and 4.90 (2015) mg TE/g DW. -Total phenolic content: Average 8.00% (2014) and 3.33% (2015) higher compared to the Sole F. Total flavonoid contents: On average, 32.4% (2014) and 23.8% (2015) higher than in seeds harvested from Sole F. -Flavonoid compound content Vitexin content was higher on average by 40.2% (2014) and 17.5% (2015) than for seeds from Sole F. Isovitexin content was on average 14.9% (2014) and 9.88% (2015) higher than in Sole F. Orientin content was higher on average by 23.1% (2014) and 15.5% (2015) compared to Sole F. -Flavonoid compound content Vitexin content was higher on average by 40.2% (2014) and 17.5% (2015) than for seeds from Sole F. Isovitexin content was on average 14.9% (2014) and 9.88% (2015) higher than in Sole F. Orientin content was higher on average by 23.1% (2014) and 15.5% (2015) compared to Sole F.	------------------------	Iran	[71]
Tomato and basil, cabbage plants	Two systems compared with commercial control (cv. Rio Grande): LI: a system involving application of cow manure and manual weed control. LIMI; the same system, integrated (LI) with mulching and intercropping (basil and cabbage plants). Both systems were used to cultivate tomato line 392 harboring the hp-2 gene, which increases the pigments of plant and fruit; and tomato line 446 with the *atv* and *Aft* genes, which influence the content of polyphenols.	The LI system showed a higher content of polyphenols (+37.9%) and anthocyanins (+116.7%) in the peel and a higher content of vitamin C (+44.0%) and polyphenols (+11.1) in the pulp.	------------------------	Italy	[37]
Salicornia europaea and tomato	The experimental design forecasted three different kinds of plots, namely Salicornia in monoculture (S) (double rows of twenty-five plants each), Salicornia consociated with tomato plants (S-T) (two rows of thirteen tomato plants each, with twenty-five Salicornia plants planted at each side of the two tomato rows, and tomato in monoculture.	The cultivation method (intercropping–monoculture) had no effect on the concentration of fatty acids, chlorophylls, carotenoids, glycine betaine, total phenols, or tannins, except for flavonoids, which did decrease in concentration (−26%) in intercropped plants.	------------------------	Italy	[24]

A study carried out with a maize–peanut intercrop and another maize–soybean combination compared to a maize monoculture, with and without fertilizer application, sought to determine the quality of the maize grain in terms of its starch, protein, oil, and lysine content [70]. The economic performance, the abundance of microorganisms, and the activity of various enzymes were also reviewed. The results showed different amounts of nutrients (maize grains’ protein, oil, and lysine content) depending on whether fertilizer was used, and which type (Table 1). In general, intercropped plants showed an increase in some nutrients. Finally, the use of nitrogen fertilizer did not substantially affect the intercropping outcome of maize grains’ starch, protein, and lysine content.

A two-year experiment in China (Table 1) revealed that grain protein content may be improved by wheat and faba bean intercropping [69]. In this case, an improvement was found not only in percentage grain protein content, but also in the quantities of non-essential and essential amino acids, under different nitrogen input conditions; another study conducted in three locations in the Czech Republic (Prague, Uhříněves and Zvíkov), also investigated the grain protein content of wheat intercropped with Egyptian clover, crimson clover, red clover, white clover, common pea, dun pea, common vetch, bird’s-foot-trefoil, common kidney vetch, and fenugreek with two different intercropping methods (mixture and row-by-row cropping) [72] (Table 1). The crude protein content of spring wheat cropped with legumes was higher by 12%.

Another intercropping system involving cereal and legumes is the milpa, which is a polyculture system in which mainly maize (*Zea mays*), beans (*Phaseolus* spp.), and pumpkin (*Cucurbita* spp.) are grown together in different topological arrangements and different associated species, depending on the region. This system has been analyzed for various ecological and yield purposes [40,75]. However, in recent years, more attention has been paid to this food production system as a critical source of food and nutritional security because it provides both macro-nutrients (fat, protein, starch) and micro-nutrients (vitamins and minerals) [76]. Moreover, many milpa studies examine food yields from a different perspective; for example, a survey carried out in North America with the Iroquois group defined the quantities of energy (12.25 × 106 kcal/ha) and protein (349 kg/ha) produced per unit land area, comparing them with monoculture crops or mixtures of monocultures planted in the same area [77]. Furthermore, beyond yield and calories, other studies have shown that milpa systems produced significantly more essential nutrients. Similar results were found in a Mayan milpa system, where researchers measured the agricultural produce and nutritional content of all plants harvested from a traditional Lacandon milpa [76]. They found that, for an average family size of five individuals, this system can meet most United States Food and Drug Administration (FDA) daily value nutritional requirements per capita of calories, fat, carbohydrates, fiber, protein, vitamins A and C, calcium, iron, zinc, and niacin. In the western highlands of Guatemala, one of the world’s poorest regions, where food insecurity and malnutrition affect more than half of its inhabitants [78], a study was conducted to calculate the potential number of people fed (PNPF) considering the essential components of human nutrition, the nutrient concentrations in the common edible parts of the raw crops, and the amounts of each crop produced.

Moreover, the maize–bean–faba, maize–potatoes, and maize–bean–potatoes associations presented the highest Potential Nutrient Adequacy (PNA), contributing the most carbohydrates, proteins, zinc, iron, calcium, potassium, folate, thiamin, riboflavin, vitamin B6, niacin and vitamin C [79]. With a growing interest in beneficial substances beyond human nutrition, some review and research articles have recently been published on the bioactive compounds involved in milpa systems. These studies analyzed the bioactive and chemical composition [40,80,81] of the species involved, and others studied the nutritional and health benefits of milpa system seeds assessed in preclinical and clinical trials [82].

All these studies are related to the nutrients that are available through this system in general, since more species are involved; but there is increasing interest in studying the possible changes in the production of bioactive compounds like alkaloids, terpenoids, phenolic compounds, steroids, and other nutrients through interactions between the species in the milpa system. These compounds have been found to have bio-functional qualities: anti-inflammatory, antiproliferative, antimicrobial, antibacterial, antifungal, and anticancer properties that may be useful in therapeutic applications for human health [83]. There is particular interest in the milpa, as the most popular polyculture system in Mesoamerican countries. In 2017, a milpa of colored corn, climbing bean, squash, and tobacco with potato, three classes of peppers—poblano, jalapeno and bell—beetroot, carrot, and kale (Table 1), was planted to study its capacity to integrate bioactive-enriched vegetables that might be useful in restoring group health among American Indians, due to the growing prevalence of non-communicable chronic diseases (NCDs), such as type 2 diabetes (T2D), in this community [74]. Since the purpose was to seek a positive impact on the health of this community, tests were carried out to review the anti-hyperglycemic and anti-hypertensive properties of the vegetables used. In all cases, kale was found to have the highest α-amylase and ACE inhibitory activity, related to the quantity and quality of phenolic acids and total antioxidant activity (ABTS and DPPH).

### 3.2. Intercropping Cereals with Herbaceous Plants

A study of cereal intercropped with wheat and clover (Table 1) found higher amounts of grain protein, and the experiment also compared fertilized and unfertilized systems [73]. A higher grain protein was obtained with an unfertilized system in which wheat was sown in paired rows with broadcast clover. Another experiment studied the effect of intercropping fenugreek and buckwheat on the trigonelline content, antioxidant activity measured with 2,2-Diphenyl-1-Picrylhydrazyl (DPPH) and ferric-reducing antioxidant power (FRAP), total phenolic content, total flavonoid content, and the specific flavonoid contents of fenugreek seeds (Table 1) [84]. One of the intercropped treatments enhanced the antioxidant activity and the content of bioactive compounds, and, in general, fenugreek seeds that were intercropped with buckwheat (organic fertilizer) showed enhanced seed content of antioxidants and flavonoids. The authors explained that the increase in antioxidant activity could be caused by the overall promotion of organic manure in supplying the macro- and micro-nutrients responsible for antioxidant activity.

In other research, barley and alfalfa intercropping was combined with beneficial microorganisms, arbuscular mycorrhizal fungi (AMF), and plant-growth-promoting rhizobacteria (PGPR) (Table 1) both to improve crop yield and soil health and to increase some nutrients [71]. In this context, favorable barley yield results were obtained, and the soil’s nitrogen and phosphorus contents improved. Moreover, there was also an increase in the protein, total phenolic, and flavonoid content of the barley grains.

### 3.3. Intercropping without Legumes and Cereals

There are other intercrops involving neither legumes nor cereals, which were studied for their capacity to modify the amount of bioactive compounds in plants. Not all studies obtained favorable results regarding the production of these compounds, however, and it is important to review the reasons for this. In a study of intercropped tomato, basil and cabbage plants, yield and bioactive compounds were analyzed in the pulp and peel of the tomato fruits [37]. Here, intercropping did not have a positive effect on the production of bioactive compounds (Table 1), and only the non-intercropped system showed an increase in bioactive compounds. The authors related these results to the biotic and abiotic stress between the species.

Another intercropping system combining tomatoes with *Salicornia europaea* (Table 1) evaluates the nutritional profile and the content of some health-promoting compounds of the edible portion of *Salicornia europaea* [24]. Additionally, the antioxidant, antibacterial, and anti-inflammatory properties of *S. europaea* were studied to characterize its bioactivity. The cultivation method did not affect nutrients and bioactive compounds, except for flavonoids, where the intercropping system’s content decreased. Nonetheless, the methanol extract from *S. europaea*, which has a potential protective effect against tumor necrosis factor (TNF), and the presence of bioactive components like cyanine, isoflavones, flavanones, etc., indicate that antioxidant, anti-inflammatory activity and neuroprotective properties can be developed in this intercropping system.

## 4. Another Approach for Improving Nutritional Profile in Intercropping Systems

Sustainable agriculture can encompass various agricultural techniques, significantly affecting food production. This is a latent concern, since these systems must balance human needs with respect for natural resources [85].

In the arduous task of finding the ideal conditions for multiple cropping systems, a number of studies have been conducted, and researchers have already collected valuable information. A review in 2015 gathered several studies about the biotic interactions that could take place in different polyculture systems to provide more ecosystem services, and also offered information on how to implement a polyculture step by step based on other research [18]. These guidelines for designing multiple cropping systems combine ecological, agricultural, and genetic concepts and approaches. A similar review in 2021 emphasized the aspects that must be considered when implementing intercropping strategies for the purpose of enhancing food and environmental security. This study also highlighted the importance of different factors, such as the choice of crops and cultivars, sown proportions, and agronomic management, including water and nutrients [14]. Neither review supported a particular approach to change or ameliorate the nutritional profile of the crops for human benefit. However, both studies pointed to one latent problem where part of the research can be focused on improving the nutritional profile of the edible parts in this type of system: the availability of the trait values of cultivars and specific eco-physiological models for the adequate construction of an ideotype.

An adequate construction of the ideotype (biological model that is expected to perform in a predictable manner within a defined environment or conditions) is necessary to ensure the crops’ competitive capacity, as well as identifying the different complementarity and facilitation processes of soils in intercropped systems by using key species to reduce inter-specific competition, and recognizing the key role of soil microorganisms [86]. The balance between these characteristics affects yield and promotes ecosystem efficiency, which are the overarching objectives of intercropping systems [14,18]. Another crucial area of research is plant immune systems. Relating the metabolic pathways associated with producing secondary metabolites to the type of stress that activates them is vital. Both the adaptive (eustress) and non-adaptive (distress) response to stress can significantly influence the effective channeling of plant energy into biomass production or the bioactive profile [87,88]. There is still much to study: the adaptive responses related to crop allelopathy [89]; plant–soil interactions due to vegetation patterns around the plants of interest [90]; where the different species used enrich the microbiota; the plant environment where species belonging to a region can interact and fit together better [91]; the plant–plant interaction [92]; and where physio-agronomic parameters are sought (cereals with legumes). Furthermore, the exact point at which the different qualities (improved yield, improved nutritional profile, soil health, etc.) are manifested in each crop study has been little studied at present, which is why the different types of stress exerted by these interactions are at the forefront of the current research in any type of food production system.

## 5. Materials and Methods

The articles in this review were found using keywords like food security, intercropping, bioactive compounds, grain quality, nutrient profile, sustainable agriculture, human health, eco-physiology, and phytochemical properties. Ninety-two research and review articles were consulted for this document, using different sources (MDPI, Taylor and Francis, Research Gate, Springer, Google Academic, Cross reference, Science Direct).

More specifically, nine articles were classified into one table. This table summarizes the results of investigations into intercropped systems that aimed to modify the nutritional profile of one or more species involved in human nutrition. According to the methodology that was implemented, the studies were classified according to the types of nutrients being improved: Table 1 shows research in which there was an improvement in the quantities and quality of both bioactive compounds and macronutrients, or one of these characteristics.

The studies that did not meet these conditions were not considered for development in this portion of the document. However, some are mentioned in other sections because they contain critical points regarding the adequate development of intercropped systems that seek to improve food security.

## 6. Conclusions and Future Perspectives

To the best of our knowledge, this is the first study to survey the research carried out to date on intercropped systems introduced for the purpose of modifying nutrients and bioactive compounds in the cultivated species in order to improve human health. Some of these investigations obtained favorable results in modifying the amounts of bioactive compounds, some macronutrients, or even both. In contrast, other studies did not find an impact on these indicators regarding the different treatments implemented in intercropping.

Although one of the main objectives of SDG goal 2, “zero hunger,” is to improve the quality of food for the benefit of human consumption through agroecological systems, researchers must still fill in various gaps. In general, the research related to nutritional profiles in intercropped systems for human health needs to be more broadly based, al-though there may be more studies related to this topic available in other sources. We did find articles in which the type of species is repeated, but there is no clear explanation for why such an association was implemented, nor of the physiology or allelopathic interactions. Further research is necessary into how we might influence the metabolic pathways of interest in order to produce specific bioactive compounds and be able to test their bioactivity on human health. To this end, the content of the bioactive compounds of several species grown in a monoculture is being reviewed more effectively to obtain a nutraceutical benefit in humans [40,82,93], and these studies could pave the way to obtaining a more complete understanding of metabolic pathways and the different types of stress that activate them. This could be helpful for comparing and improving the amount of these bioactive compounds or macronutrients in intercropped systems under other conditions. In these studies, a possible change in some organoleptic characteristics, such as the (bitter) taste of foods in which bioactive compounds have been increased [94], has been mentioned. This characteristic suggests a wide array of research topics that may arise involving polyculture systems.

Likewise, the yield and environmental benefits associated with intercropped systems should be considered, even if the primary goal is to improve human nutrition. Most of the studies that were reviewed did not neglect these characteristics, since there is enough information to support the environmental benefits of polycultures and the measurable improvement in yield. Agroecological techniques have even been improved by adding different types of fertilizers and microorganisms, which improve soil health and nutrient management [4,70,83]. These same techniques should also be studied with a focus on their effect on the phytochemical profile of the plants.

Another area where existing research is scarce and further study would be useful is the importance of regional species in intercropping systems. Some studies have proven their effectiveness in reestablishing biodiversity and natural habitats according to their level of adaptation to the different ecoclimatic conditions of the areas in which they are located [95,96]. There are even some articles that found a development of nutraceutical properties [74]. These are very valuable areas of study, since the development of a sustainable diet requires that the most significant possible benefits from different socioeconomic and environmental perspectives are obtained without losing the culinary and cultural customs of the region, which is important for food security worldwide [57].

Finally, the question of whether we have the necessary information to develop intercropping systems to benefit human health must be answered from various perspectives. Several studies provide a basis for the implementation of intercropped systems with this objective. However, all this information must be analyzed, along with the plant ecophysiologies and immunities (hormetic response) that will interact in a polyculture, to formulate new theories, and thus construct an appropriate ideotype [85,97]. This would allow for growers to produce foods with a proper nutritional profile, and even to obtain a benefit that goes beyond conventional nutrition and improves food security and the health status of the population in general.

## Data Availability

Not applicable.

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
