# Peer review of "Intercropping Systems to Modify Bioactive Compounds and Nutrient Profiles in Plants: Do We Have Enough Information to Take This as a Strategy to Improve Food Quality? A Review"

_plants, 2024, doi:10.3390/plants13020194_

Round 1
Reviewer 1 Report
Comments and Suggestions for Authors
Review “Intercropping systems to modify bioactive compounds and nutrient profiles in plants, do we have enough information to take it as a strategy to improve food quality? A review.”
The subject proposed in review is relevant because the intercropping is little discussed in the literature in terms of nutritional quality while the issue is also important. The number of experimental trials on which the work is based is limited around 30 publications with 9 more precisely valued in tables.
Several terms are used referring to the expected qualities of plants from intercropping: food security, nutritional profile, food quality, human nutrition, phytochemicals, bioactive compounds, macronutrients. The term “food quality” is used in the title although it is addressed human nutrition with bioactive compounds in the article.
The abstract remains very synthetic by information scattered from the literature. It would however have been interesting to show which compounds among macronutrients and bioactive compounds can be favored by the association and to answer concretely the title that it could be a quality improvement strategy
Results
Line 50: the term “cultivars” does not seem adequate (talk about “systems”)
Line 69: the study mentioned is not referenced
Line 83: the phrase “has yet to be achieved (24)” lacks negation
Paragraph 2 is constructed from only part 2.1
Table 1 : indicate the percentages in relation to the control
Table 2 : it lacks the third treatment of the «Row» trial
Table 3 : it lacks the results of the effects of the cultivar modality used
Line 345-347: the sentence is redundant
Line 391: it is mentioned the processes of competition but not those of complementarity and facilitation within an intercropping
Line 405 the term “improve” is indicated and in the tables the term “modify” is used
Line 366-368: the sentence is not clear
Material and methods
Line 401: why did you choose the keyword "eco-physiology"? In the choice of key words, the different association systems known in different regions are not used for this research: cereals and vegetables, etc....
Author Response
Dear Review,
Attached please find a digital copy of the revised version of our manuscript entitled: "Intercropping systems to modify bioactive compounds and nutrient profiles in plants: Do we have enough information to take it as a strategy to improve food quality? A review." authored by Ana Patricia Arenas-Salazar, Mark Schoor, Benito Parra-Pacheco, Fernando García-Trejo, Irineo Torres-Pacheco, and Ana Angélica Feregrino-Pérez, for the publication in Plants. The manuscript has been carefully revised and modified based on the reviewers ‘comments.
Revisions made to the manuscript are marked in different colors that editors and reviewers can easily see them. Our point-by-point response to the observations is also described below.
Reviewer 1. Round 2
Thank you for the improved version of the Manuscript.
-Related comments for reviewer 1 were highlighted in yellow in the MS.
We appreciate your time and dedication in reviewing this manuscript.

Reviewer 2 Report
Comments and Suggestions for Authors
First, I want to admit that the subject of the research is very interesting with a view to future cropping techniques and potential new agricultural practice.
However, the work ist very very simplistic and generalized. I did not make any comments in the manuscript as there are a lot of aspects missing throughout the whole paper.
To get some structure into the paper, i would recommend follwing points:
1) I would recommend deciding wether to focus on subsistence farming or on large-scale farming. For example, the authors did quite a lot of research on the MILPA system - which was developed by the ancient Mayas. For substience farmers this system can have advantages but when focussing on large-scale farming at the given techniques and practices an extensive roll-out is unlikely due to missing LARGE-SCALE harvesting techniques.
2) There is no information on how intercropping effects the yield which is the most important index for farmers to be willing to try out new cropping systems. Furthermore, there are almost no statements about the feasability of the intercropping sytems as well as pros and cons in practical terms.
3) The authors provide almost no information on how the nutritional value (nutrients, secondary metabolites with health-promoting effects) of one crop can be enhanced or reduced by a second crop species in an intercropping system. For example in line 321 the authors state the positive aspects of kale. But are these positive aspects natural properties of the plant or can they be increased by intercropping? Here, I would recommend a proper literature study on the effects of plant-plant-interaction or kin-recognition (Ninkovic et al.; Dudley et al. and many more....) and how plants can affect the metabolites (primary and secondary) of their neighbors.
Further comments:
Beginning line 174 the authors state that reasearch about the interaction between agrochemicals and human health ARE BEGINNING!!!! This is really not true: only in Google Scholar thousands of studies are available. Furthermore there are also a lot of studies available of how agrochemicals can influence (positively and negatively) the primary and secondary metabolites of plants. For example: There are couple of studies availabel of how N-fertilizers can decrease the Nitrogen-uptake of legumes as rhizobia become "lazy" and ineffective (Kiers et al. 2007...). Or studies on how a moderate herbicide application can lead to an increase in secondary metabolites due to stress >> plants produce more secondary metabolites as defence mechanism when they are facing stress which can lead to an increase in sec. metabolites = increased nutritional value.
Hidden hunger: This topic is really generalized in the manuscript as it is not due to "wrong" agricultural practice but also due to the crops used. For example billions of people around the globe are completely dependent on Maniok/Cassava which is the basic food in many developing coutries but known to be poor in Vitamine A (blindness, liver deseases etc.). Hereby, I would recommend more proper literature study....
Comments on the Quality of English LanguageThe quality of the English is very poor. Many sentences are absolutely not understandable due to lack of grammar AND, in particular, nested sentences which are much too long. Furthermore authors use a lot of fillers which, all together, makes the manuscipt really hard to read.
Author Response
Dear Review 2,
Attached please find a digital copy of the revised version of our manuscript entitled: "Intercropping systems to modify bioactive compounds and nutrient profiles in plants: Do we have enough information to take it as a strategy to improve food quality? A review." authored by Ana Patricia Arenas-Salazar, Mark Schoor, Benito Parra-Pacheco, Fernando García-Trejo, Irineo Torres-Pacheco, and Ana Angélica Feregrino-Pérez, for the publication in Plants. The manuscript has been carefully revised and modified based on the reviewers ‘comments.
Revisions made to the manuscript are marked in different colors so that editors and reviewers can easily see them. Our point-by-point response to the observations is also described below.
Reviewer 2.
Thank you for the improved version of the Manuscript.
-Related comments for reviewer 2 were highlighted in green in the MS.
We appreciate your time and dedication in reviewing this manuscript.

Reviewer 3 Report
Comments and Suggestions for Authors
Reviewers comments
Overall I think that this is an interesting topic and worthy of research interest, but the presentation of this review could be vastly improved. Throughout, there is a tendency to use long sentences of paragraphs, which often did not flow well together. Parts of the review seemed to be off topic. The presentation of the methodology was quite unscientific (see how to present this for a systematic review) and I am not convinced that the search criteria or method was fully inclusive to the topic. More specific comments:
Abstract
Line 12- ‘seem to be’ related is a strange way of saying it, maybe just correlated with? It sounds a bit more certain
Line 13- consider rephrasing to ‘this type of food production system leads to the loss…’ and change the ‘had’. I’d have thought for this is it ok to say it in the present tense because it is an ongoing problem but this makes it sound more historic
It might be worth giving a brief definition of intercropping in the abstract
Line 19- ‘plant nutrient composition’ not nutrient plant composition.
Consider splitting some of these sentences up to make them shorter and more simple, particularly the last sentence and sentence beginning line 18
Main text
Again consider splitting up some of the longer sentences. I also expected to see a definition of intercropping given
Line 50- what do you mean by ‘these cultivars’ should this read ‘these cultivation systems’?
Line 56- is it just endemic species or all species?
Line 65- what do you mean by ‘finish products’
Line 100- this dimension cannot be accomplished..? or not accomplished with current food systems maybe
As interesting as some of the content is, I find section 2 (before 2.1) to be a bit unrelated to the topic and with a long review as it is, it could be better to condense this section down and simply focus more on the nutrition aspect.
Consider splitting the first paragraph of section 2.1. none of it flows particularly well.
I don’t get the connection between the sentences on line 155- they seem like 2 completely different arguments but appear to follow on
Section 3 is written much better than the sections above it. The sentences flow well together and they are much more simple to follow.
Line 208- nutritional content? Not species
Line 223- this paragraph seems out of place- you mention studies that do it but instead of describing them it is more like a discussion of them. I would expect you to first review these different studies/ systems first and then analyse them.
The way that the studies are presented could be improved. Firstly, the reference to each study should be presented at the end of the first sentence that discusses it, not at the end of the section. The phrasing of these sections could be more informative to focus more on the nutritional content and result as opposed to the cropping mix. I also feel like maybe some studies are missing, there must be more studies than presented here that look at cereal- legume systems and influence on protein, N or P content. Maybe a systematic review approach would be better to give a comprehensive overview of the topic. [edit] I saw that this was accomplished using a systematic review approach. Maybe that could be more clear from the start
Line 324- if this is the only study, then why are more studies presented in the table?
Line 356- basil cabbage? Or tomato, basil and cabbage?
Line 358- what do you mean by ‘did not produce good results’. Do you mean did not positively improve content?
Line 367- define TNF
Line 390- you may need to define ideotype for those unfamiliar with the concept
Comments on the Quality of English Language
The language itself is mostly good although in parts I detected typos and grammatical errors. But throughout the sentences could be reduced in length and simplified.
Author Response
Dear Review,
Attached please find a digital copy of the revised version of our manuscript entitled: "Intercropping systems to modify bioactive compounds and nutrient profiles in plants: Do we have enough information to take it as a strategy to improve food quality? A review." authored by Ana Patricia Arenas-Salazar, Mark Schoor, Benito Parra-Pacheco, Fernando García-Trejo, Irineo Torres-Pacheco, and Ana Angélica Feregrino-Pérez, for the publication in Plants. The manuscript has been carefully revised and modified based on the reviewers ‘comments.
Revisions made to the manuscript are marked in different colors so that editors and reviewers can easily see them. Our point-by-point response to the observations is also described below.
Reviewer 3.
Thank you for the improved version of the Manuscript.
-Related comments for reviewer 3 were highlighted in blue in the MS.
We appreciate your time and dedication in reviewing this manuscript.

Round 2
Reviewer 2 Report
Comments and Suggestions for Authors
The manuscript has been improved substantially and is from my point of view now ready for publication.
Author Response
We appreciate your comment and thank you for your time and dedication in reviewing this manuscript.
Reviewer 3 Report
Comments and Suggestions for Authors
Overall I think that this version does read much better than the previous version so well done with that. I still believe that it would be improved by removing section 2.1 completely and focussing on the nutritional aspect of the system. As it stands, it is almost like you start the paper, have that section then effectively restart it again. Whilst it is disheartening to lose a chunk (which can be recycled to another paper anyway), it would improve this manuscript to lose it.
line 358- are appropriately managed (not by)
line 836- typo in depend(s)
I also disagree with your response to the comment about presenting the studies. It is the common acceptable method to cite the study at first mention, not wait til the end. The majority of these sections describe the method as opposed to the interesting results and impact on nutritional content.
For improved flow I think combining the 3 tables together into 1 encompassing all the studies would improve the manuscript. This would immediately highlight all studies performed on intercropping for nutrition and would mean the text can be grouped better.
line 1274 repetition of methanol
line 1293- 'or amelioration of' makes no sense here
plant parameters is not an adequate definition of the ideotype
Comments on the Quality of English Language
The language is much better. there are some typos which could be corrected. Still some very long sentences and the paragraph structuring is also not great in parts.
Author Response
We appreciate your time and dedication in reviewing this manuscript.
